# Role of Creative Therapies in Gynecological Oncology: Results of a Multigenerational Survey in Patients and Caregivers

**DOI:** 10.3390/cancers16030599

**Published:** 2024-01-31

**Authors:** Bettina Jantke, Jalid Sehouli, Matthias Rose, Jolijn Boer, Andreas Jantke, Desislava Dimitrova, Hannah Woopen, Adak Pirmorady-Sehouli

**Affiliations:** 1Kinderwunschärzte Berlin, Center for Sterility Treatment and Fertility Protection, 14195 Berlin, Germany; 2Medical Department, Section of Psychosomatic Medicine, Corporate Member of Freie Universität Berlin and Humboldt-Universität zu Berlin, Charité—Universitätsmedizin Berlin, 10117 Berlin, Germany; 3Department of Gynecology with Center for Oncological Surgery, Corporate Member of Freie Universität Berlin and Humboldt-Universität zu Berlin, Charité—Universitätsmedizin Berlin, 13353 Berlin, Germany; jalid.sehouli@charite.de (J.S.);; 4North-Eastern German Society of Gynecological Oncology (NOGGO), 13359 Berlin, Germany; 5European Guild for Medicine and Culture (EUKMK), 10827 Berlin, Germany

**Keywords:** creative therapies, art therapy, creative writing, music therapy, medical care, activating unconscious, cancer treatment, supporting treatment

## Abstract

**Simple Summary:**

Creative therapies like painting, dancing, and writing are often suggested to support the treatment of illnesses, severe illnesses, including cancer, but there are not many detailed studies on how well they work in hospitals. We asked women with cancer, their female family members, and female hospital workers about how creative therapies could make cancer treatment better. Out of 718 people who answered, most tried innovative therapies to feel better and be healthier. Many liked writing as a way to help themselves. The answers show that creative activities could help us understand how patients move, feel, and interact with others. Even though many patients do not usually talk to their doctors about their hobbies, many would like to include creative activities in their medical care. Also, people prefer to perform these creative activities in groups, meaning hospitals should think about organizing such group activities to help patients.

**Abstract:**

Introduction: Although creative therapies like painting, dancing, and writing are often used and encouraged to treat various diseases, including cancer, there are few systematic scientific studies on innovative therapies in medical care. Methods: An anonymous survey was developed for female patients, their relatives, and female medical staff on the impact of creative therapies on optimizing clinical therapy management in exclusively female trials. Results: Of 718 respondents, 358 were female patients, 69 were medical personnel, and 291 were in the control group. Overall, 91.2% of respondents had sought access to creative therapies, indicating strong self-motivated engagement in activities to improve health and well-being. This study also uncovered a significant preference for creative writing among patients. Furthermore, the data suggest that integrating innovative therapies into biopsychosocial anamnesis could offer valuable insights into patients’ mobility, mood, and social behaviors. Despite a general hesitation to discuss leisure activities with medical professionals, many patients wanted to incorporate creative activities into their treatment plans. Moreover, group settings for innovative therapy were preferred, highlighting the need for more structured support in medical environments to facilitate these therapeutic interactions. Conclusions: This study suggests creative therapies can be valuable in medical care.

## 1. Introduction

Depression and anxiety in patients with cancer are a common problem during and after treatment and are very stressful for patients. Watts et al. were able to show that the prevalence of anxiety and depression is significantly higher in patients with ovarian cancer than in the healthy female population [1]. This is particularly important given the increasing significance of improving survival management and quality of life [2]. In this context, creative therapies are often used in treating various diseases and promoted in cancer guidelines. Still, few systematic scientific studies exist on using innovative therapies in medical care.

In modern medicine, there is an increasing understanding that healing is not achieved exclusively by medication and traditional therapy methods. Creative therapy is a field in which artistic expression is combined with psychotherapy to achieve healing, well-being, and personal change [3]. This established method can significantly improve quality of life and well-being, especially when therapists have a solid education and professional standards as a quality characteristic. Our survey is intended to investigate the different groups concerning this topic and to show the other uses of creative therapy by patients, medical staff, and a young, healthy control group. The unique feature here is the experience in the patient group about an activity started before the illness. Most studies only focus on using creative therapies as an additional application during therapy to improve health and quality of life.

There is a lack of studies on the type of stimulation and organization and the timing of the start of activities already carried out with patients. This may reveal opportunities to reach patients and staff better and promote implementation. It is well known that patients benefit from creative therapy during cancer therapy. Our study intends to show whether the performance of an activity that has already begun changes in connection with a disease and the reasons for this. This can offer an approach to address this specifically and to eliminate possible causes such as high costs. It should also be shown how consciously the test subjects use an activity to improve their well-being in the first place and how actively they wish this to be used for disease management, perhaps also for relatives.

Last but not least, the question of the doctor–patient relationship also plays a significant role here, as our study is also intended to find out whether there is a conversation or the desire for a discussion with the treating doctor and whether the doctor can play a vital role as the initiator of creative activity. This, in turn, can be helpful in the further course of the illness and improve the patient’s well-being and state of health.

Art, dance, and music therapy are essential parts of 21st-century integrative therapy and are now present in all areas of healthcare and the treatment of most psychological and physiological conditions. Pratt et al. reported as early as 2004 that these therapies would play a significant role in 21st-century healthcare practice as they contribute to the humanization and comfort of the modern healthcare facility [4].

Art therapy plays a vital role here in coping with stress, anxiety, and pain, both in supporting the healing of patients and in compensating for stress on staff in nursing professions. Our study will also examine how medical staff would like to use coping strategies, how they are already using them, and how they can best be integrated into everyday working life.

In interventional oncology, integrative medicine can help reduce the fear of pain before and after treatment [5].

Gynecological cancers account for around 14.4% of newly diagnosed cancers and are a significant cause of mortality and morbidity in women. Although conventional therapeutic methods have improved average life expectancy, they also profoundly affect patients’ physical and psychological experience, negatively impacting their quality of life during the disease-free interval. To achieve improvements with additive measures such as creative therapies, measurements such as the HRQoL (Health-related Quality of Life) are now carried out before and after treatment. Gil-Ibanez et al. already compiled an essential summary of these measurement instruments and the historical context, which outlines the current knowledge of the effects of gynecological cancers and their therapies on patients’ quality of life [6].

A review by Acher et al. summarized four articles from 2015 that demonstrated how cancer patients who received various creative interventions during therapy encountered a reduction in anxiety and depression, as well as improvements in quality of life, stress management, anger, and mood [7].

In creative therapies, such as the process-based use of innovative processes like painting or music and dancing, movement, or writing, there are numerous additional creative expressions in the therapeutic process.

Mindfulness-based interventions, in particular, can also lead to a reduction in anxiety, depression, fatigue, and stress in cancer patients and thus lead to an improved quality of life, as Xulin et al. showed in a systematic review in 2019 [8].

Many of these methods have already been used in holistic treatment and rehabilitation, not only for female patients. Music therapy, for example, was already investigated in 2017 by Aalbers et al. in a meta-analysis of patients with depression, which showed effects on depressive symptoms and reduced anxiety, leading to improved patient well-being [9].

All these therapies develop individual expression and the interaction between body, soul, and emotions. They also offer a way to manage stress and promote emotional relief and physical rehabilitation, thereby contributing to recovery. Creative therapies can lead to significant improvements in quality of life, health status, and well-being in the context of disease management, especially in patients with chronic diseases who, at times, also undergo long-term treatments such as cancer diseases. The conduct of creative therapies in groups can have an additional positive effect on individual reflection and the processing of emotions. The experiences of the affected and the social interaction, as well as the exchange among the group members, can reduce the feeling of isolation and allow the aspect of the “universality of suffering” to be experienced in the sense of Yalom, which can often be supportive in chronic diseases and prolonged medical treatments [10]. However, work-related psychological stress can also occur in members of all healthcare professions and is widespread. In 2020, Moss et al. published a study on four creative art therapy programs to improve mental stress and turnover intention among health professionals with burnout symptoms [11]. Medical staff, especially nurses, have always been and continue to be exposed to a great deal of stress, ranging from long and irregular working hours to dealing with pain, loss, and emotional suffering and also caring for relatives. In addition to these factors, additional aspects include staff shortages, financial constraints on companies, more complex patients, and changing technologies. The associated stress can severely impair the cognitive functions, attention, and memory of those affected, leading to severe errors in the treatment of patients. Therefore, it is essential to involve medical staff in programs that improve the mindfulness of those affected to reduce stress [12].

In 2023, Engel et al. reviewed these interventions based on art therapy and showed that empathy, connection, and tolerance of ambiguity could be promoted and that this can positively affect burnout [13].

The research in the present paper shows to what extent patients are already using these additional creative therapies, considering Charité gynecology department medical staff and a group of patients from a fertility practice as a control group, and how their application can be further optimized. It also highlights the extent to which patients and staff demand integrating such therapies into standardized treatments for optimal results for the affected patients.

## 2. Materials and Methods

In the context of this scientific survey, the following section explains the methods and results used to document the research process properly.

Given the previous presentation of the theoretical aspects and the empirical hypotheses, the research design of the present work will now be presented in greater detail. For this purpose, the design as such is described first to illustrate the operationalization based on it. Subsequently, the sampling will be broken down. Finally, the statistical evaluation procedures will be clarified.

### 2.1. Procedure and Participants

The present work is an empirical analysis in cross-section design. In an anonymous survey, information was asked about the impact of creative therapies on the optimization of clinical therapy management in exclusively female trials. For this purpose, a semi-structured questionnaire was developed in an inter-professional and interdisciplinary working group, including a statistician, which consisted of 23 questions and was written in German. To validate the questionnaire based on a pilot study, the questionnaire was tested on ten female patients for readability and understandability. This was useful to verify if a few patients understood the questionnaire. We carried out a pretest and a test statistical analysis before the questionnaire was revised and issued. A quantitative study design was chosen because it offers the advantage of generating many observations quickly and thus provides the most efficient way of data extraction. Before, it was validated by an ethics committee. To ensure anonymity, the questionnaires were not numbered. They were handed out by hand to the test subjects, who, after completing the questionnaires, placed them in a locked box on the ward in the outpatient clinic or the practice. In this way, there was no possibility of tracing the individual forms.

This study was conducted by the Department of Gynecology with Center of Oncological Surgery and the Department of Psychosomatic of the Charité and officially supported by the Working group of the North-Eastern German Society of Gynecological Oncology (NOGGO) and the European Guild for Medicine and Culture (EUKMK).

Participants were recruited at the Gynecology Clinic, Charité Campus Virchow Klinikum (CVK), Berlin, Germany, and for the control group, we chose a fertility clinic, “Kinderwunschärzte Berlin” in Berlin-Zehlendorf, Germany, to compare the results with a healthy group. The participants included female patients, their female relatives, and female medical staff such as doctors, nurses, and medical assistants. From June 2022 to December 2022, 756 questionnaires were distributed exclusively to female subjects over 18, with a response rate of 720 surveys. Regarding the inclusion criteria, the survey was not bound to any conditions except age and gender. It cannot be ruled out that wrong answers were chosen because women in trials do not always respond truthfully due to external influences or social preferences.

Only unanswered questionnaires and completed ones of women under 18 were excluded. The final sample size included *n* = 718 female participants. We divided the respondents into three main groups:Patients;Medical personnel;Control group.

The group of patients included outpatients and patients undergoing treatment in the Gynecology Clinic at CVK at the time of the survey. The CVK medical staff group had doctors, nurses, and medical assistants. We selected as a control group from both locations patients who classified themselves as healthy from the clientele of a fertility clinic and the doctors and medical assistants working there, as well as relatives who described themselves as beneficial. This study was conducted in compliance with the declaration of Helsinki and was approved by the Ethics Committee of the Charité—University Medicine Berlin, Germany (Research Ethics Committee Reference Number: EA4/021/19).

### 2.2. Statistics

The dependent variable, health status or stress level, represents the “health status and stress level” scale, which contains ten metrically scaled items (see Table 1). An open question to the subjects determined the independent variable of age and belonging to the three main groups. All the questions to the participants were partially nominal, quasi-metric, or ordinal scaled.

A one-factor ANOVA was used as a statistical test to determine whether a correlation was significant. Finally, in the case of a significant ANOVA, post hoc tests such as Bonferroni or Games–Howell were used to show the difference between the variables. The choice of the post hoc test depended on the previously calculated variance homogeneity. The prerequisites, such as the standard distribution assumption using the Shapiro–Wilk test, the absence of outliers by graphical representation of box plots, and the variance homogeneity using the Levene test, were always carried out beforehand.

We also used the *t*-test to compare the means of the two groups. The dependent variable was metrically scaled, and the independent variable had more than two categories. The prerequisites for the *t*-test were also the Shapiro–Wilk test, the absence of outliers due to the graphical representation using boxplots, and the Levene statistic.

Furthermore, we used the Chi2 test to determine the correlation between two variables. The requirement was no less than five observations per cell. Otherwise, the test was interpreted according to Fisher. Both the independent and the dependent variables were nominally scaled.

## 3. Results

### 3.1. Study Participants

Of the 720 female participants in this study, 718 could be assigned to a group: patients, medical staff, or control group. Two respondents provided no information, so they were excluded from the analysis. Of the 718 respondents, 358 were female patients (49.9%), 69 were medical personnel (9.6%), and 291 were in the control group (40.5%). The youngest surveyed was 18 years old, and the oldest was 88 years old. The median age of all participants was 39 years, and the mean was around 44 years. When looking at the subgroups, the median age of the patients, the medical staff, and the control group was 51 years, 33 years, and 35 years, respectively (see Table 2).

### 3.2. Disease Entity

In the female patient group, over two-fifths (41.3%) had a malignant disease, around half (49.2%) had a benign disease, and less than one-tenth (6.7%) had endometriosis (see Table 2). Ten female participants did not report their illnesses. Of the 148 patients with malignant disease, 109 (73.6%) had primary disease and 39 (26.4%) had recurrent disease. The group of patients reported a lower self-assessed health status (M = 6.13, SD = 2.526) than the group of medical staff (M = 8.5, SD = 1.327) and the control group (M = 8.33, SD = 1.127).

The health status of those with primary disease (M = 4.95) was, based on a *t*-test, also significantly better than that of those with recurrent disease (M = 3.95) *p* < 0.010.

### 3.3. Health Status

The majority of the female participants, 71.6%, reported feeling healthy. Only 28.4% of female participants reported being ill.

The participants were also asked about their subjective health status assessment on a scale from 1 (low) to 10 (high). Based on the mean (7.17 points) and the median (8.0 points) of the assessed health status, this was indicated as relatively high. The self-assessed health status of the patients was significantly worse than that of the medical staff and the control group. The self-assessed health status of the control group did not differ from that of the medical staff. An ANOVA was calculated, which was significant at *p* > 0.01.

### 3.4. Stress Level

The participants were asked about their subjective assessment of their stress levels on a scale from 1 (low) to 10 (high). The self-assessed stress level averaged 5.49 points and a median of 6.0 points when considering all groups. It was further investigated whether the stress level in the group of female patients differed depending on the type of illness compared to the control group. An ANOVA was also calculated in this case. A significant correlation was calculated with *p* > 0.19. Using the Bonferroni post hoc test with equal variances, a substantial difference in self-assessed health status was found between the control group and the group of malignant patients and between the group of malignant and comorbid patients and the group of benign patients and the control group. When looking at the descriptive statistics, the malignant disease group showed a mean value of M = 4.68, the benign disease group showed a mean value of M = 7.22, and the control group showed a mean value of M = 8.36. Pearson r was calculated to determine the correlation between age, stress level, and health status.

The stress level of the primary sufferers (M = 5.75) was, based on a *t*-test, significantly higher than that of the recurrent sufferers (M = 5.05) *p* < 0.044.

The result of the correlation measure showed that two variables, namely, age and health status, correlate moderately negatively (r = −0.434, *p* ≤ 0.001), and stress level and health status correlate weakly negatively (r = −120, *p* ≤ 0.001). The substantive conclusion is that the higher the age, the worse the health status; the worse the health status, the higher the stress level.

An examination of the correlation between self-assessed stress levels and changes in activity after an illness using ANOVA showed a significant result of *p* > 0.04 (See Figure 1).

The Bonferroni post hoc test then showed a significant difference in the self-assessed stress level between the participants who had responded with less activity after illness and those who had responded with the same. When looking at the descriptive parameters, the less activity group showed a significantly higher stress level with M = 5.95 than the same activity group with M = 5.22.

### 3.5. Activity for Health

Most participants (*n* = 655; 91.2%) regularly engage in activities to improve their health, quality of life, and well-being; less than one-tenth (*n* = 63; 8.8%) reported not engaging in activities to improve their health status. A Chi2 test was carried out and showed no significant difference (*p* = 0.922) between patients, staff, and the control group. The cross-tabulated values showed that the group of patients (91.6%), the group of medical staff (91.3%), and the control group (90.7%) use creative therapies with approximately the same frequency. In the group of malignant patients, the Chi2 test also showed no significant difference with *p* = 0.299 regarding creative therapies.

### 3.6. Reported Activities for Health

The survey also asked what type of activity the respondents perform, and 15% of the participants indicated painting, 13% creative writing, 70% exercise, 42% music, 2% clay work, 15% meditation, only 1% soapstone work, and 27% dancing. Overall, 27% reported other activities such as praying, singing, cooking/baking, handicrafts, reading, and photography.

An ANOVA (analysis of variance) was executed to examine whether the age of patients impacted the implementation of creative therapy. It should be noted that a normal distribution was previously tested but not reported. Nonetheless, the ANOVA is considered a robust procedure for data that is not normally distributed and has a sample size of at least 30. A box plot was subsequently used to test for standard deviation, which was provided, and to assess variance homogeneity using Levene’s statistics (F(1) = 0.003, *p* = 0.955). The results of the ANOVA indicated non-significance (*p* = 0.652). As demonstrated by the descriptive statistics, the average age of individuals who underwent creative therapy was 43.60 years, while those who did not undergo therapy had an average age of 42.65 years. A Chi2 test was conducted to check the correlation between the primary group variable and the variables including painting, writing, movement, sound work, meditation, music, and dancing. A significant correlation with the three main groups (patients, medical staff, and control group) was only calculated for creative writing activity (*p* = 0.016). It can be concluded that 16.2% of female patients use creative writing most frequently. In the medical staff group, only 7.2% used this activity, and in the control group, 9.6% of the respondents used this activity. Regarding the subdivision of the group of malignant patients into primary and recurrent patients, a significant result was only found for exercise in the Chi2 test *p* = 0.033. When comparing the mean values, 78.2% of those with primary tumors reported exercise as a creative activity, and only 21.8% of the recurrent patients reported this activity.

### 3.7. Organization of the Activities

Most respondents (69.5%) organized their activities privately, 22.8% in an association, 17% in both, and only 1% in a medical institution. A Chi2 test was calculated to check the correlation between the creative activity’s organizational form and the three main groups. Six hundred forty-eight cases were examined, and 76 were excluded as no information was provided. The Chi2 test with *p* = 0.538 was not significant in this study. Thus, no correlation could be found in the group of patients, the group of medical staff, or the control group about the organizational form of an activity. The cross table shows that only 1.8% of the patients surveyed carry out creative activity in a medical facility.

In the group of patients with malignant disease, no significant difference was found between the primary and relapsed patients, *p* = 0.185. The only striking aspect when comparing the cross-tabulation was that none of the patients in the relapsed group were organized in a medical facility, compared with 2% of the primary patients.

### 3.8. Motivation for an Activity to Improve Health

The incentive to pursue an activity to improve health, quality of life, or well-being was discovered by themselves in 49.9% of the respondents, in 17.8% by friends, in 13.1% by family, in 4.5% of cases by medical rehabilitation, and in only 2.4% of cases through a clinical stay. The Chi2 test was used to investigate the correlation between the three main groups and the stimulus that led to the performance of creative activity. A significant correlation, *p* < 0.001, was calculated. When looking at the cross table, it can be seen that the stimulation by friends, family, and self-discovered activity was similarly distributed across all three main groups. However, encouragement from medical rehab or the clinic was significantly higher in the group of female patients (8.6%) than in the medical staff (5%) and the control group (0.3%). The Chi2 test was calculated for the group of malignant and benign patients and the control group to further investigate creative therapy’s stimulation, which was also significantly positive, X2(16) = 42.668, *p* < 0.001, V = 0.150. Based on the contrasting values in the cross table, it can be seen that the group of malignant and benign patients showed a high level of expression about the “self-discovered” group. The group of malignant patients showed a higher level of concern regarding the suggestion of medical rehabilitation.

### 3.9. Beginning of an Activity

Most respondents, 38%, reported engaging in activity since childhood, and 35.1% had not begun one until adulthood. Only 6% started an activity after illness, and 9.6% could not precisely remember the motivation. An ANOVA was calculated to determine the relationship between the items when the respondents used an activity to improve their well-being and self-assessed health status. The post hoc test, according to Bonferroni, showed a significant difference in health status between the group who had started an activity in childhood or adulthood and the respondents who had only begun this activity after becoming ill.

When looking at the mean values of the descriptive statistics, it can be seen that better health status was indicated if the activity was started in childhood (M = 7.38), or at least before the illness in adulthood (M = 7.28), than if the activity was started after an illness (M = 5.48).

### 3.10. Frequency of Activity

An important question was how an activity already carried out had changed due to an illness. The survey determined how often the participants engaged in a creative activity. Most participants (42.2%) engaged in weekly activity, 33.7% participated in improving their health, quality of life, and well-being daily, 4.7% performed this monthly, and 8.8% said they seldom partook in one. Of all respondents, 10.6%, thus 76 participants, did not provide any information. The survey also investigated the reasons for infrequent implementation. These reasons were lack of energy, lack of talent, lack of motivation, lack of time, and high costs. A Chi2 test was performed to calculate the correlation between the variables: how often an activity was completed and why it was rarely performed. The test was significant for the reason of lack of energy, time, and motivation, with *p* < 0.001; for the reason of cost, with *p* = 0.010; and for the statement lack of talent, with *p* = 0.013. In summary, it was calculated that when an activity was rarely carried out, the reasons given were lack of energy (41.3%), lack of motivation (22.2%), lack of time (42.9%), lack of talent (4.8%), and cost (4.8%).

### 3.11. Changed Activity Due to a Disease

One-fifth (20.3%) of respondents stated that the frequency of engaging in an activity did not vary due to a disease, 15.6% indicated that they participated less frequently due to a disease, and 12.5% reported that participation increased due to an illness. Since the variable changed activity due to an illness is a Likert scale and thus, a quasi/metric agreement scale, an ANOVA was calculated for further verification to show a difference concerning the variable changed activity and the variable main groups with the group’s control group, medical staff, and patients. There was no significant difference between the groups (F(2) = 0.184, *p* = 0.832). No significant difference (*p* = 0.469) could be determined by calculating an ANOVA, even when considering the group of malignant patients in primary and recurrent patients.

There was a significant difference between the stress level and health depending on the frequency of the activity performed. Using ANOVA, it was shown that the test subjects who performed an activity less frequently had a higher stress level than those who performed an activity immediately or more after the onset of an illness (*p* = 0.044). Regarding the stated health status of the test subjects, the ANOVA also showed a significant result with *p* < 0.001 if patients responded with less activity after an illness; this correlated with significantly lower health status (M = 6.96 to M = 5.80)

When participants responded to the questionnaire with the answer “seldom”, the reason was lack of energy in 12%, lack of time in 10%, lack of motivation in 6%, and lack of artistic talent in 1%, as well as associated costs in 1%. Overall, 2% of the respondents marked “other reasons” in the survey. Reasons given included not yet completed recovery and severe fatigue. Since the survey was handed out during the Coronavirus pandemic, the participants also indicated, among these other reasons, that the pandemic prevented them from completing an activity. Pain, finding housing, and uncertainty due to acute illness were also listed. A Chi2 test was used to investigate whether powerlessness, lack of talent, motivation, costs, or lack of time were related to the patient, medical staff, and control groups. A significant correlation with the patient group was only found for powerlessness, with *p* < 0.001. The *t*-test showed no significant result when testing a correlation between primary tumor and recurrent tumor for the reason of lack of activity.

### 3.12. The Link between Health and Activity

The respondents were asked whether they saw a connection between their state of health and the creative activities they carried out, and 45% of the women surveyed saw a connection between partaking in an activity and their health condition; 27.7% marked “absolutely” in the questionnaire. Only 18.9% saw no correlation between health, quality of life, and well-being and creative activities (see Table 3).

We also investigated whether individual disease entities, such as benign diseases, malignant diseases, and people living with endometriosis, saw a connection with the performance of activity regarding their health, quality of life, and well-being compared to the control group. A Chi2 test was also calculated for this investigation, which was significant at X2(8) = 24.467, *p* = 0.002, V = 0.136. When looking at the cross table, it was clear that the malignant disease group showed a higher expression level than the other groups because there was no connection. Thus, this group also showed a lower level of complete agreement with seeing a connection between activity and health than the comparison groups. In this respect, the Chi2 test did not show any significant correlation when examining the group of primary tumor patients and recurrence patients (*p* = 0.821).

### 3.13. Activity and Talking with Doctors or Employers

A total of 66.7% of the respondents stated that at the time of the survey, they had not talked with their doctor about engaging in an activity to improve their health.

Only 16% had already discussed this with their doctor. Only 35.7% of the respondents would have liked to have addressed an activity with their attending physician, while 48.5% were uninterested.

A Chi2 test was calculated to examine the correlation between the control group, the medical staff, and the group of patients. This was weakly significant at X2(2) = 7.517, V = 0.112

When looking at the compared values using the cross table, it can be seen that the group of patients had a higher level of desire to discuss activities with a doctor than the group of medical staff and the control group. In contrast, the medical staff and the control groups did not want to discuss activities with their doctor. A Chi2 test was calculated to investigate whether the desire to talk to a doctor about the respondents’ activity to improve their well-being, quality of life, and health also depends on disease etiology. This was significant at X2(4) = 17.988, *p* < 0.001, V = 0.173.

Based on the cross-tabulated values, it was evident that the group of malignant disease patients and endometriosis patients had a stronger desire for a doctor’s consultation than the other groups. No significant differences were found in the group of patients with a primary tumor and recurrence.

The need to discuss engaging in an activity with their employer was reported by 13.2%, while 74.2% were not interested. It was investigated whether the groups of patients desired to discuss their activity with their doctor compared to the group of medical staff or the control group.

### 3.14. Integration of an Activity into Daily Work

The majority of respondents, 51.8%, would like their activity integrated into their daily work routine. Of these, 13.8% and 38.2% answered “absolutely” and “yes” respectively. Overall, 35.4% did not want an activity integrated into their daily work routine. We also investigated whether the respondents would like to incorporate their activity into their everyday working lives. For this purpose, a Chi2 test was used to calculate the correlation between the group of patients, medical staff, and the control group and the variable activity/workday. This correlation was not significant (X2(4) = 3.324, *p* = 0.505).

Of the participants surveyed, 82.1% were interested in integrating creative therapies to cope with the illness. Of these, 26.9% answered “Absolutely”, and 55.2% answered “Yes”. Of the respondents, 9.3% had no interest. The aim was to investigate further the extent to which the respondents in the three groups—patients, medical staff, and control group—would like to be offered creative ways for coping with illness.

For this purpose, a Chi2 test was used to calculate the correlation between the three groups and the variable desire for coping with illness. No significant correlation was found (X2(4) = 6.164, *p* = 0.187). We also found no significance between patients with primary tumors and recurrent patients (X2(2) = 0.964, *p* = 0.617).

### 3.15. Activity Offers to Relatives

Most respondents would also like creative activities offered to family members to cope with the illness. Overall, 14% of the participants responded, “Absolutely”, while 55.6% replied “Yes”, and 20.2% were “not interested” in such an offer for relatives.

A Chi2 test was also conducted to assess how the participants’ wishes were distributed between the patients, the medical staff, and the control group. This result showed a significant correlation (X2(4) = 22.896, *p* < 0.001, V = 0.133).

Based on the compared values in the cross table, it was evident that the female patients showed a higher degree of not wishing to have their relatives involved.

On the other hand, the group of medical staff showed a greater desire for the involvement of relatives through a creative offer. It was then further investigated whether there was a difference in the group of patients divided into those with benign and malignant disease, endometriosis patients, and the control group according to a desire for a creative offer to relatives as part of coping with the disease. A Chi2 test was calculated for this investigation, which was significant (Chi2(8) = 50.269, *p* < 0.001, V = 0.197). Based on the compared values in the cross table, it can be seen that the group of endometriosis patients shows a higher expression and the group of patients with malignant disease shows a lower expression concerning the desire for a creative offer to relatives as part of coping with the disease.

### 3.16. Guidance for an Activity

Most respondents, 52%, would like guidance in a group setting, 37% as individual coaching, 34% through an app, 31% as a book, and 12% as a blog. Of the respondents, 2% indicated other preferences. The need for a space for the activity to take place at the ward, in addition to workshops, video formats, and Zoom meetings, was also mentioned.

A Chi2 test was conducted to show a correlation between the individual forms of instruction of creative activity and the patient group, the medical staff, and the control group. Only one significant result was found for instruction in app form (X2(2) = 10.330, *p* = 0.006, V = 0.120) when looking at the cross-tabulation; the medical staff group showed a significantly higher preference for using an app than the other groups.

## 4. Discussion

Creative therapies in medical care have received increased attention in recent years as they have shown clinical and psychological benefits for patients in previous studies. Patients with tumors have a significantly increased risk of anxiety, depression, and sadness compared with the average population, as Pitman et al. showed [14].

Mental and physical health, in particular, can be improved by processing emotions. For example, physical sensations are consciously perceived, accepted, and expressed through art therapy, and cancer patients, in particular, benefit from this [15].

These art therapies could potentially help reduce symptoms of anxiety as well as depression and improve the quality of life in adult cancer patients. However, further research with stringent methods is needed due to heterogeneity in the interventions and the limited methodological quality of studies [16].

The malignant disease group showed a significant not-to-see connection between health and activity. We also saw no significant correlation when examining the group of primary tumor and recurrence patients.

This is particularly interesting as the infrequent performance of creative activity was associated with poorer self-rated health and higher self-rated stress levels in our survey, and we saw a connection with rare activity after the onset of the disease as being caused by a lack of strength in the group of patients.

In a 2023 study, Valero-Cantero et al. showed that family caregivers also benefit from music therapy [17]. Our results showed a desire for relatives to be involved in a creative program for coping with illness. Most respondents would also like creative activities offered to family members to cope with the disease. Interestingly enough, and statistically significant, the female patients showed a higher degree of not wishing to have their relatives involved. On the other hand, the group of medical staff showed a greater desire for the involvement of relatives through a creative offer. It was then further investigated whether there was a difference in the group of patients divided into those with benign and malignant disease, endometriosis patients, and the control group according to a desire for a creative offer to relatives as part of coping with the disease. The group of endometriosis patients showed a higher expression, and the group of patients with malignant disease had a lower expression concerning the desire for a creative offer to relatives as part of coping with the disease.

In a 2020 review, Raybin et al. suggested that a connection can be established with creative expression, that coping is facilitated with creative activities, that communication is enhanced with innovative interventions, and that continuity, the concept of time, can only be experienced through creative expression [18].

In 2017, Lee et al. published a study indicating that cancer-related anxiety and depression were significantly improved and the prevailing severe anxiety and depression during cancer treatment were reduced considerably with art therapy, i.e., painting [19].

This study showed a significant correlation with creative writing in the group of female patients of the Charité gynecology department. In our research, the female patients used creative writing significantly more often than the medical staff and control groups; we found no significant difference in the malignant patient group with regard to primary or recurrent disease, which was also shown by Zhu et al. in a study [20], where creative writing in special classes led to an improvement in a patient’s mood.

According to a survey by Vergo et al., it made a difference whether writing was simply expressive or guided by a writing coach [21]. While uncomplicated, expressive writing showed no positive influence, guided writing with a trained coach positively influenced patients with terminal cancer [21].

Joly et al. did not find a significant improvement in severe fatigue syndrome in cancer patients one month after radiotherapy by complementing it with art therapy [22]. However, a positive influence on social well-being and motivation was found.

A systematic overview work by Bradt et al. in 2021 also showed that music interventions can have a positive effect on anxiety, depression, hope, pain, and fatigue in adults with cancer compared with standard care [23]. This could also have a positive impact on children with cancer when used concomitantly with respective therapies.

Our study shows which possibilities in the sense of a biopsychosocial anamnesis are currently overlooked in our daily clinical work. Asking patients about creative activities within the framework of an active and stable doctor–patient relationship is a valuable benefit of our study, as it allows for making statements about our patients’ mobility, mood, and social behavior. In a 2022 review, Akeep et al. also confirmed the importance of good communication between doctors and patients in oncology to enable evidence-based and patient-centered care [24].

However, only 13.7% of our respondents wanted to discuss a creative activity as part of their treatment after a doctor’s interview. We found that the group of patients had a significantly higher desire to discuss this activity with a doctor than the group of medical staff and the control group. We also found that the desire and willingness to talk to a doctor about the respondents’ activity to improve their well-being, quality of life, and health statistically significantly depends on disease etiology.

The group of malignant disease patients and endometriosis patients had a stronger desire for a doctor’s consultation than the other groups. No significant differences were found in the group of patients with primary tumors and recurrence.

A significant difference was shown between creative therapies and the studied individual groups, including the medical staff, female patients, and the control group.

However, our study showed that the fewest activities were practiced in an association, i.e., in a group, and that activities were most frequently organized privately (69.5%). Only 2.8% were completed in an association, i.e., a group, and even less were completed in a medical institution (1%). Since engaging in activities in a group and a medical institution with other affected persons could contribute to improving well-being and quality of life, supporting this form of organization would be beneficial.

Lagattolla et al. 2023 showed a significant decrease in anxiety levels in an individual music therapy group [25]. In contrast, the integrated music therapy group had a higher perception of the help received and the use of personal resources.

In our study, the group setting was followed by individual coaching; self-guidance, for example, with a book; and digital instruction with apps and blogs. Only the medical staff group showed a significantly higher preference for using an app than the other groups. In a 2023 review by Saevarsdottir et al., the increasing importance of digital media as support during the active treatment phase on breast cancer patient’s quality of life and well-being was confirmed [26].

Since most of the participants in our study discovered creative therapy through their initiative, followed by friends and family, and only 4.5% through medical rehabilitation, more specifically, 2.4% through the clinic in which they were treated, one starting point would be to promote access to creative therapy, especially for patients in a medical facility. As a further step, this would also be well-suited for special medical staff groups, especially in clinics with chronically ill patients where the psychological burden is high to be able to maintain health and reduce stress.

Art therapy, in particular, could prevent emotional exhaustion and psychosocial problems for medical staff, as shown in systemic reviews by Tjasink et al. in 2023 [27] and Philipps et al. in 2019 [28].

When used by certified therapists, which is a sign of quality, creative therapies can generally be a valuable and essential addition to conventional therapies [29]. These therapies can be diverse and include newer therapies such as aromatherapy or equine therapy in addition to the traditional established methods, which can support the healing process of patients. In a 2022 study, Czakert et al. showed that aromatherapy, in particular, can promote the well-being and mindfulness of patients [30].

In this context, studies on aromatherapy with essential oils conducted by Corasaniti et al. led to a reduction in the intensity of pain in cancer-related pain, especially in the final stages of the disease [31]. Inhaled aromatherapy, in particular, appears to be beneficial here. For example, Nascimento et al. conducted a meta-analysis of different pain conditions [32] and showed that the type of oil, the type of pain, and the time of inhalation were essential variables.

Most studies that deal with improving the quality of life of oncology patients primarily examine music therapy and painting therapy. In our study, we demonstrated a significant correlation between the patients at Charité and the use of creative writing to improve their quality of life. In contrast to other studies, our study investigated creative activity during an illness, its effect on the quality of life, and whether the patients had already started a creative activity to improve their well-being earlier, regardless of illness. The environment in which the activity was practiced was also examined. Most studies only investigate how a creative activity, specifically in therapy, improves quality of life. Our study also analyzes the extent to which patients wish to integrate the activity into their daily therapy routine and discuss this with medical staff.

In contrast to other studies, we were able to show how much potential lies in biopsychosocial anamnesis. Compared with other studies, our study investigated how a particular therapy improves the quality of life during oncological treatment, for example, and which creative activity the participant’s subjects already bring to the treatment. Our study also showed that the organizational form of innovative therapy is relevant to the patients and that organization in a group is desired.

There are several limitations to this study. This study is a one-site evaluation of different time points of malignant and non-malignant diseases. Additionally, the potential impact of the individual patient has been not analyzed; therefore, future studies should evaluate the attitude of the medical staff and the access to creative therapies. Furthermore, the questionnaire was only provided in German, thus excluding patients who do not speak German, and future studies should provide questionnaires in multiple languages. Nevertheless, based on our large study and high compliance rate, we believe that this study provides relevant information for the clinical routine and scientific community.

Future studies should further explore the extent to which specific therapies can most effectively support particular patient groups. It would also be helpful to look at the long-term effects of creative therapies to show whether they provide sustainable benefits in terms of disease management.

## 5. Conclusions

In summary, the findings of this study suggest that creative therapies can be a valuable tool in medical care. Overall, it is clear that innovative therapies are an essential tool in medical care, not only for patients but also for medical staff. The creative therapy approach contributes to emotional support and stress reduction for patients and offers medical staff space for reflection and coping. The multiple benefits include improved communication, stress reduction, improved health status, promotion of team dynamics with group guidance, and a holistic enrichment of the healthcare system. This study is intended to describe the current situation and thus illustrate the relevance of creative therapies in medical treatments. Oncological patients, in particular, should be able to benefit from these forms of therapy as recommended in the German guidelines. Still, the preventive use of creative therapies should also be considered, so it is important for us to present a broad field of application. A new finding of this study is that there is an active resource in our patients that is not sufficiently asked about and used for the treatments, although it can be helpful and supportive.

However, more studies should be completed to promote and explore their integration into clinical everyday life.

## Figures and Tables

**Figure 1 cancers-16-00599-f001:**
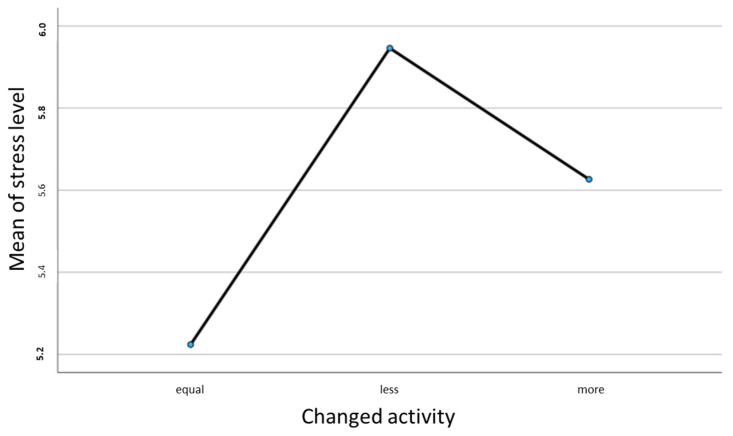
Correlation between the mean value of the self-assessed stress level and the changes in activity.

**Table 1 cancers-16-00599-t001:** Operationalization table.

Hypothesis	Variable	Item	Survey
1	Dependent	Stress level/health status	10 items on a 10-point Likert-scale (1 = strongly disagree, 10 = strongly agree) Mean score
1	Independent	Age/group	Open question

**Table 2 cancers-16-00599-t002:** Participant characteristics.

	Total (*n* = 718)	Patients (*n* = 358)	Medical Staff (*n* = 69)	Control Group (*n* = 291)
Age				
M (SD)	43.5 (16.0)	50.1 (17.5)	35.0 (12.5)	37.4 (10.5)
MDN	39	51	33	35
Min–Max	18–88	18–88	18–59	18–78
Sex				
Female, *n* (%)	718 (100)	358 (100)	69 (100)	291 (100)
Disease entity				
Malignant, *n* (%)	-	148 (41.3)	-	-
Benign, *n* (%)		200 (55.9)		
Primary tumor, *n* (%)	-	108 (15.2)	-	-
Recurrent tumor, *n* (%)	-	39 (5.4)	-	-

M, mean; SD, standard deviation; MDN median.

**Table 3 cancers-16-00599-t003:** The link between health and activity.

Response	*n* (%)
Yes, absolutely	199 (27.7%)
Yes, partly	323 (45.0%)
No	136 (18.9%)
No response	60 (8.4%)

## Data Availability

The data presented in this study are available on request from the corresponding author. The data are not publicly available due to privacy and ethical reasons.

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
