# Peer review of "Role of Creative Therapies in Gynecological Oncology: Results of a Multigenerational Survey in Patients and Caregivers"

_cancers, 2024, doi:10.3390/cancers16030599_

Round 1
Reviewer 1 Report
Comments and Suggestions for Authors
The authors designed the research “Role of Creative Therapies in Gynecological Oncology: Results of a Multigenerational Survey in Patients and Caregivers” by including 718 female participants. . In the female patient group, over two-fifths (41.3%) had a malignant disease, around half (49.2%) had a benign disease, and less than one-tenth (6.7%) had endometriosis. I suggest the authors focused the malignant patients or analyzed subgroup of malignant patients instead of patients with large variation.
Comments on the Quality of English LanguageMinor editing of English language required
Author Response
Thank you very much for your positive feedback and your valuable comments that we have incorporated into the article. Please find our point-by-point response as follows:
Pages 5, 6, 8, 9, 10, 11,12,13
You are correct that a focus in our study exclusively on carcinoma patients would also have been of great interest especially since the journal is primarily concerned with studies on malignant diseases.
Therefore, we focused more precisely on cancer patients we includes patients with endometriosis because this disease is of special interest and endometriosis as associated disease (e.g.) endometrioid and clear cell ovarian cancer and endometrial cancer.
This review is intended to describe the current situation and thus illustrate the relevance of creative therapies in medical treatments. Oncological patients, in particular, should be able to benefit from these forms of therapy as recommended in the German guidelines. Still, the preventive use of creative therapies should also be considered, so it is important for us to present a broad field of application, but we have implemented some evaluations that only refer to carcinomas such as primary tumor and recurrence
Regarding the revision of the English language: Thank you for this tip, we have had the paper corrected again by a native English speaker.

Reviewer 2 Report
Comments and Suggestions for Authors
In principle, this study is interesting, although the explanation of the project is very limited.
Moreover, the evaluation presented in this cross-sectional article with its questionnaire and its limited/descriptive statistical analysis, makes me think that this article has a scientific and evidence-based level too low to be published in a high-impact journal like cancer.
I would advise the authors to resubmit their article to another MDPI journal, but with a slightly lower impact factor.
Author Response
Thank you very much for your positive feedback and your valuable comments that we have incorporated into the article. Please find our point-by-point response as follows:
Manuscript: Page 5 line 204 – 218, Pages: 6-11
Thank you very much for the valuable advice that mentioning only the descriptive statistics in the results section is insufficient for a journal like Cancers. Therefore, we additional explorative statistical analysis.
Interdisciplinary networking is a necessary prerequisite for good medicine; it is important that areas that are difficult to measure also try to become scientifically visible to expand medicine by a perspective that is difficult to visualize but no less important.

Reviewer 3 Report
Comments and Suggestions for Authors
Thank you for the opportunity to review the manuscript. The following are my suggestions to help enhance the quality of the manuscript:
The first statement in the abstract lacks proper punctuation, leading to confusion among readers about the clarity of the statement. Even in the abstract, the expression of the English language is improper, and I recommend that the authors undergo proper English language editing. This issue persists throughout the entire manuscript.
In the introduction section, some paragraphs are too short, consisting of just one sentence. Authors are advised to enhance these paragraphs. Additionally, in the introduction section, the literature on the use of creative art therapies is not well-documented, and the rationale for conducting the current study is not presented. Authors are encouraged to identify the gap in the current research and provide the rationale for conducting the study.
While reviewing the 'materials and methods' section, the opening sentence sounds inappropriate for scientific writing. The paper, overall, requires language editing.
Please explain how the sampling procedure ensures anonymity.
Provide details about the statement, 'The dependent variable, health status or stress level, is represented by the "health status and stress level" scale, which contains 10 items and is metrically scaled,' with appropriate references.
Provide details on how dependent and independent variables were coded. The lack of information on variable coding makes it difficult to comprehend the results.
In the discussion section, a word suddenly appears in capital letters, giving readers the impression that thorough language editing is needed. Additionally, some paragraphs in the discussion section consists of only one sentence. Author(s) are advised to strengthen the discussion in the paragraph.
Explain more about the limitations of the current study and provide strengths before the conclusion.
Overall, the current study is in draft form and requires proper organization.
Author Response
Thank you very much for your positive feedback and your valuable comments that we have incorporated into the article. Please find our point-by-point response as follows:
Authors reply:
Thank you for the opportunity to review the manuscript. The following are my suggestions to help enhance the quality of the manuscript:
- In the introduction section, some paragraphs are too short, consisting of just one sentence. Authors are advised to enhance these paragraphs. Additionally, the literature on using creative art therapies is not well-documented in the introduction section, and the rationale for conducting the current study is not presented. Authors are encouraged to identify the gap in the current research and provide the rationale for conducting the study.
Authors reply: Page 2 lines 60-79,87-89
The full manuscript underwent an intransitive editing and revision by a native speaker.
- Please explain how the sampling procedure ensures anonymity.
Authors reply: Page 4 lines 160 - 164
As requested by the data safety and ethical review board this study this study was conducted with participant anonymity to minimize potential additional bias.
- While reviewing the 'materials and methods' section, the opening sentence sounds inappropriate for scientific writing.
The paper, overall, requires language editing.
Authors reply:
Thank you for the comment we revised the chapter intensely and deleted this opening sentence.
- Provide details about the statement, 'The dependent variable, health status or stress level, is represented by the "health status and stress level" scale, which contains ten items and is metrically scaled,' with appropriate references
Authors reply: Page 5 lines 193 - 203
In the methods section, we have also explained the health status and stress level scales in more detail and included the dependent and independent variables for a better understanding of the results.
- In the discussion section, a word suddenly appears in capital letters, giving readers the impression that thorough language editing is needed. Additionally, some paragraphs in the discussion section consists of only one sentence. Author(s) are advised to strengthen the discussion in the paragraph.
Authors reply:
We would also like to thank you for your valuable advice on streamlining the paragraphs in the discussion section. We have strengthened the discussion in the paragraph and had the paper corrected again by a native English speaker.
- Explain more about the limitations of the current study and provide strengths before the conclusion
Authors reply: Page 14 lines 623 - 631
We have also expanded the section before the conclusion and pointed out the study's limitations and strengths. “ There are several limitations of the study. The study is a one-site evaluation of different time points of malignant and non-malignant diseases. Additionally the potential impact of the individual patient have been not analyzed therefore future studies should although evaluate the attitude of the medical stuff and the access to creative therapies. Furthermore, the questionnaire was only provided in German, thus excluding patients who do not speak German, and future studies should provide the questionnaire in multiple languages. Nevertheless based on our large study and high compliance rate we believe that this study provides relevant information for the clinical routine and scientific community.“
- The current study is in draft form and requires proper organization.
Finally, we would like to thank you for your important and valuable advice, and by implementing your tips, we have brought the study into a more appropriate organizational form.

Reviewer 4 Report
Comments and Suggestions for Authors
Dear author’s,
I was pleased to review your article and i have the following comment’s:
Improving quality of life in patients with gynecologic cancer is highly necessary in our practice, so the study is interesting.
It is mandatory to give details in methods about the questionnaire. It was validated? It was in English?
Study limitation are mandatory to be highlighted in the section discussion.
Please explain the novelty of your study. What new info brings to the existing literature?
Minor punctuation edits.
Author Response
Thank you for your positive feedback and the valuable comments we have incorporated into the article. Please find our point-by-point response as follows:
- It is mandatory to give details about the questionnaire's methods. Was it validated? Was it in English?
Authors reply: Manuscript: Page 4 lines 152 – 158, 162-172
The questionnaire was developed based on a comprehensive literature review and interdisciplinary and interprofessional workshop including patients so the questionnaire has the goal to reflect the need to creative therapies the questionnaire was not designed to become a validated tool, therefore no systematic validation process was required. Nevertheless, the evaluation of understanding and readability in 10 patients was tested in advance. These aspects have been reported more precisely in the manuscript.
- Study limitations are mandatory and are to be highlighted in the section discussion.
Authors reply. Page 14 lines 627-634
There are several limitations of the study. The study is a one-site evaluation of different time points of malignant and non-malignant diseases. Furthermore the potential impact of the individual patient have been not analyzed therefore future studies should although evaluate the attitude of the medical stuff and the access to creative therapies. Additionally, the questionnaire was only provided in German, thus excluding patients who do not speak German, and future studies should provide the questionnaire in multiple languages. Nevertheless based on our large study and high compliance rate we believe that this study provides relevant information for the clinical routine and scientific community.
- Please explain the novelty of your study. What new info brings to the existing literature?
Authors reply: Page 13, lines 607 – 622 Page 13 lines 610-613, page 14 lines 614-625
We have now clarified more precisely our results and their clinical and scientific implications. In contrast to various other trails, we have not focused on specific type of creative therapies. Additionally we could observe the high use of creative therapies in both cohorts of woman with malignant diseases and benign diseases. In our opinion, the observation can be used as an additional source to strengthen self-confidence and resistance. Additionally the majority of patients with malignancy diseases demand the integration of creative therapies within their cancer management.
Our study also analyzes the extent to which patients wish to integrate the activity into their daily therapy routine and discuss this with medical staff. In contrast to other studies, we were able to show how much potential lies in biopsychosocial anamnesis. Compared to other studies, it investigated how a particular therapy improves the quality of life during oncological treatment, for example, and which creative activity the participant's subjects already bring to the treatment. Our study also showed that the organizational form of innovative therapy is relevant for the patients and that organization in a group is desired.
A new finding of the study is that there is an active resource in our patients that is not sufficiently asked about and used for the treatments, although it can be helpful and supportive.
- Minor punctuation edits.
Authors reply:
We have also improved the punctuation.

Round 2
Reviewer 2 Report
Comments and Suggestions for Authors
The authors have considerably increased the quality of their article. Bravo, it looks much better this way
Reviewer 3 Report
Comments and Suggestions for Authors
I can see that the author(s) have addressed the reviewer's concern, and the paper is much improved/
Reviewer 4 Report
Comments and Suggestions for Authors
Thank you for your response.